# Theoretical modeling of QCD radiation in off-shell Higgs production through gluon fusion

**Rafael Coelho Lopes de Sá**[1][⋆], **Martina Javurkova**[1,2][†], **Matteo Lazzeretti**[3][‡] **and Raoul Röntsch**[3,4][◦]

**1** Physics Department, University of Massachusetts Amherst, Amherst, Massachusetts, USA
**2** Physics Department, Matej Bel University, Tajovského 40, Banská Bystrica, 97401, Slovakia
**3** Tif Lab, Department of Physics, University of Milan and INFN, Sezione di Milano, Via Celoria 16, I-20133 Milano, Italy
**4** INFN, Sezione di Milano

⋆ rclsa@umass.edu , † martina.javurkova@cern.ch , ‡ matteo.lazzeretti@studenti.unimi.it , ◦ raoul.rontsch@unimi.it

## Abstract

The measurement of the Higgs boson width is a critical test of the Standard Model, with significant implications for understanding electroweak symmetry breaking. Direct measurements are limited by detector resolution, but it can be measured with greater precision through a combined analysis of on-shell and off-shell Higgs boson production. While results for on-shell production have been computed to a very high accuracy, theoretical predictions for off-shell Higgs boson production are not as well controlled due to the breakdown of the heavy-top approximation and the large interference with non-resonant amplitudes. Seeking to understand and improve the theoretical control, we compare leading-order and next-to-leading-order plus parton shower differential cross-sections for signal, background, and full physical processes in off-shell Higgs boson production at the Large Hadron Collider, using POWHEG, MADGRAPH, and SHERPA. We analyze the impact of higher-order quantum chromodynamics effects and theoretical uncertainties, highlighting differences between predictions using jet merging with parton showers, and those from next-to-leading order computations matched to parton showers. The results provide insights for improving theoretical predictions and their application to experimental measurements in the future.

# 1 Introduction

The Higgs boson plays a central role in the Standard Model (SM), with its properties being crucial to understanding the mechanism of electroweak symmetry breaking. The Higgs boson width is a particularly important parameter, as it encapsulates key information about the particle's interactions and its lifetime. While a direct measurement of the width from the line shape is limited by the experimental mass resolution, it can be extracted with significantly improved precision through a combined analysis of on-shell and off-shell Higgs boson production [1–3]. This is possible because, for Higgs bosons produced via gluon fusion and subsequently decaying into a pair of massive electroweak bosons, $gg \to H \to VV$, a sizeable quantity of Higgs bosons are produced off the mass shell [4]. Other production modes such as vector boson fusion also yield significant rates of off-shell Higgs bosons and can be exploited to obtain complementary bounds on the Higgs boson width [5].

Theoretically modeling off-shell Higgs boson production in gluon fusion faces two obstacles. First, gluon fusion is a loop-induced process that proceeds predominantly through a top-quark loop. While theoretical predictions for on-shell Higgs boson production can often be simplified by considering the top-quark mass as infinite (the so-called heavy top limit), such an approximation does not hold in the off-shell regime, where the virtuality of the Higgs boson can be comparable to or even exceed the top-quark mass. Thus, at leading order (LO), the computation requires a one-loop amplitude with full top-quark mass dependence, while the next-to-leading order (NLO) correction necessitates a two-loop amplitude. Second, in this region of phase space, large destructive interference effects occur between the Higgs boson signal and the continuum background process, $gg \to VV$. As the latter is a loop-induced $2 \to 2$ process, the calculation of its amplitude beyond LO in QCD is extremely challenging, particularly when considering massive quarks in the virtual loops. Indeed, while the two-loop amplitude for the process $gg \to H$ was computed many years ago [6–10], the equivalent amplitude for the background process with massive virtual quarks was only computed relatively recently [11–13], and the first fully NLO-accurate results for off-shell Higgs boson production were only published last year [14]. Previous calculations of the off-shell production cross-section and differential distributions to NLO in QCD have relied on the large top-quark mass expansion [15–17] or a reweighting of massless two-loop amplitudes [18, 19] to evaluate massive two-loop amplitudes.[1] In Ref. [24], the calculation of Ref. [16] was matched to parton showers in the POWHEG formalism [25–27] using the POWHEG-BOX-RES package [28], and the resulting gg4l code was made public.[2] Results with next-to-leading order accuracy

---

[1]The two-loop massive amplitudes have also been computed using a high-energy expansion [20], and using a combination of the large-top mass, threshold [21] and high-energy expansions using Padé approximants [22]. A recent computation using a complementary expansion in small transverse momentum has also been shown to agree very well with the exact amplitudes [23].

[2]The code can be downloaded in the POWHEG-BOX-RES directory from https://powhegbox.mib.infn.it.

matched to parton showers (NLO+PS) were also presented in Ref. [29], which also includes the matching of the $pp \to ZZ$ process to parton showers at NNLO accuracy using the MiNNLO$_{\text{PS}}$ method [30, 31]. Increasing the fixed-order accuracy for offshell Higgs production while including the background and interference effects is not feasible at present. Indeed, obtaining NNLO accuracy would require the computation of $gg \to ZZ$ at three loops and (e.g.) $gg \to ZZ + g$ at two loops, both of which are beyond our current capabilities. The latter amplitude would also be required for an NLO calculation in the presence of a hard jet. Jet-merged calculations can therefore only be obtained at LO accuracy, and are possible for off-shell Higgs boson production in gluon fusion using MADGRAPH [32], and 0- and 1-jet merging is also available in SHERPA [33, 34], making use of OPENLOOPS libraries [35, 36].

Based on the comparison between off-shell and on-shell Higgs boson production, both the ATLAS [37, 38] and CMS [39, 40] collaborations have performed analyses of the Higgs boson width using the $ZZ$ leptonic decay channels using the LHC Run 2 data set, yielding values of $4.3^{+2.7}_{-1.9}$ MeV and $3.2^{+2.4}_{-1.7}$ MeV, respectively. These results are consistent with the SM prediction of 4.1 MeV [41]. For these measurements, CMS generated samples at LO with MCFM 7.0.1 [2, 42]. In contrast, ATLAS generated separate samples for the signal and background processes, as well as the signal, background and interference (SBI) combination using SHERPA with OPENLOOPS at LO with up to one additional jet merged (0+1j). Both experiments normalized their samples based on the next-to-leading-order (NLO) cross-section calculated in Ref. [17] and an estimate of the next-to-next-to-leading order and next-to-next-to-next-to-leading order $k$-factors calculated for on-shell Higgs bosons at $m_H = 125$ GeV [43, 44]. Complementary measurements of the Higgs boson width using the same technique have also been made in the $WW$ decay channel [45] and the production of four top quarks [46].

The experimental extractions of the Higgs boson width from off-shell studies are still limited by statistics. For example, the most recent measurement performed by the ATLAS Collaboration in the $gg \to ZZ$ decay channel [38] quotes a modeling uncertainty of 10% on the measured signal strength. This is negligible compared to the 50% statistical uncertainty from the LHC Run 2 dataset, but will become dominant with the 3000 fb$^{-1}$ of integrated luminosity projected for the HL-LHC program.

As detailed above, the theoretical modeling of off-shell Higgs production using Monte Carlo (MC) event generators – including real and virtual radiation at the matrix element level, parton showers (PS), and their consistent matching – poses significant challenges. Consequently, theoretical uncertainties, especially those affecting jet-related observables, remain a major source of systematic uncertainty across all off-shell measurements. To further understand these issues, in this paper we present a tuned comparison of three MC event generators: POWHEG + PYTHIA8, MADGRAPH + PYTHIA8, and SHERPA, ensuring that all common parameters are the same in each program. The first of these codes provides a NLO-accurate calculation matched with parton shower, while the second and third perform LO jet merging with different prescriptions. As such, the results provide an estimate of the impact of virtual QCD corrections at NLO precision as well as the size of various theoretical modeling uncertainties for off-shell Higgs studies, and allow us to assess the reliability of current MC modeling strategies. To the best of our knowledge, this is the first time that results from different event generators based on parton shower matching and merging have been compared for off-shell Higgs boson production. This investigation aims to provide a foundation for developing a more robust understanding of MC uncertainties for this process.

We organize the paper as follows. In Sec. 2, we briefly discuss the computational setup, parameter settings, and fiducial cuts. In Sec. 3, we present the differential cross-section results for the off-shell Higgs signal, continuum background, and the SBI combination, which includes interference effects. In Sec. 4, we examine the impact of systematic uncertainties arising from variations in the parameters used in the calculation. Finally, we conclude in Sec. 5.

## 2  Computational Setup

This section outlines the computational framework used for generating events in off-shell Higgs production via gluon fusion, focusing on the different MC generators employed. In Sec. 2.1, we describe the key aspects of each generator, including matrix element calculations, parton shower matching, and merging procedures. In Sec. 2.2, we provide details on the numerical configurations, fiducial cuts, and parameter settings used to ensure consistency across the generators and facilitate a reliable comparison of QCD modeling methods.

### 2.1  Event Generation Setup

We employed three MC generators to simulate events for off-shell Higgs boson production: POWHEG (through the gg4l process in POWHEG-BOX-RES), MADGRAPH5_AMC@NLO v3.5.0, and SHERPA v2.2.10. Each of the three generators employs different methods for handling QCD radiation and parton shower, which we briefly review here.

The implementation in POWHEG [24] allows the user to obtain separate results for the signal and background processes, and the SBI combination at NLO accuracy, matched to a PS. The virtual amplitude for the Higgs-boson-mediated process $gg \to H \to VV$ as well as that for the background process $gg \to VV$ with massless quark loops are included exactly, the latter using the library GGVVAMP [47]. The two-loop amplitude for the process $gg \to VV$ with massive quarks can be included either using an expansion in $1/m_t$, via reweighting, or a combination of these. For this study, we chose to use reweighting, which has been shown to reproduce the exact NLO results very closely [14]. The amplitudes for the real corrections are provided by the OPENLOOPS library, and include both gluon-initiated ($gg \to ZZ+g$) and quark-initiated ($qg \to ZZ + q$ and $q\bar{q} \to ZZ + g$) channels. We followed the default implementation in POWHEG and included contributions that have at least one vector boson attached to a closed fermion loop in our results, see the discussions in Refs. [17,27] for further details. We matched the fixed-order calculations to the PYTHIA8.309 parton shower for this study.

We used MADGRAPH to generate LO and LO+1-jet samples using its internal routines to compute the one-loop amplitudes, and then merged these samples to the PYTHIA8.309 parton shower using the MLM scheme [48, 49]. As for POWHEG, diagrams with at least one vector boson coupled to a closed fermion loop were included, and results for signal, background, and SBI were simulated.[3]

SHERPA calculates the LO and LO+1-jet amplitudes using its own implementation of the OPENLOOPS libraries. Separate results can be obtained for the signal and background processes as well as the SBI contribution, although results for the interference piece by itself cannot be simulated. As for POWHEG and MADGRAPH, amplitudes are required to have at least one vector boson attached to a closed fermion loop. The parton shower is performed using Sherpa's default parton shower tool, CSSHOWER [50–52]. The merging of parton showers with the matrix elements for multiple jet processes is performed using the CKKW-L scheme [53, 54].

Thus the three generators treat the additional real radiation along similar lines: the first emission is generated at matrix-element level, with subsequent emissions provided by the parton shower. The differences lie in the matching and merging procedures and the different parton shower algorithms. On the other hand, virtual corrections are only included in POWHEG. Understanding the complicated interplay between these distinct approaches of treating QCD radiation in off-shell Higgs predictions is the aim of this paper.

Finally, we note that merging with up to two jets has been performed in MADGRAPH [55] (albeit for stable $Z$ bosons), exploiting the automation of one-loop amplitudes in this program. This could also be possible with SHERPA if the relevant amplitudes become available in OPEN-

---

[3]We set the `auto_ptj_mjj` flag to `True` in order to generate the background and SBI sample in MADGRAPH.

LOOPS. Given the increasing importance of production through vector boson fusion in off-shell Higgs boson studies and the resulting need to model two-jet events in the off-shell regime, it would be interesting to extend the result of Ref. [55] by comparing results from the merging of up to two jets with those from NLO matched to parton shower. However, given the complexity of the two-emission amplitudes (especially for the background process), generating such samples would require substantial computing resources. We therefore defer such studies to future work.

## 2.2 Numerical Configuration

To obtain the results presented in the following sections, we considered off-shell Higgs boson production via gluon fusion in the $ZZ \to e^+e^-\mu^+\mu^-$ decay channel, ensuring that the numerical parameters and settings were consistent across all generators, including:

- Higgs boson mass: $m_H = 125.1\,\text{GeV}$ and width: $\Gamma_H = 0.00403\,\text{GeV}$;

- $Z$ boson mass: $m_Z = 91.1876\,\text{GeV}$ and width: $\Gamma_Z = 2.4952\,\text{GeV}$;

- $W$ boson mass: $m_W = 80.3980\,\text{GeV}$ and width: $\Gamma_W = 2.1054\,\text{GeV}$;

- Top quark mass: $m_t = 173.2\,\text{GeV}$ and width: $\Gamma_t = 0\,\text{GeV}$.

The relevant electroweak constants are as follows:

- Fermi constant: $G_F = 1.16639 \times 10^{-5}\,\text{GeV}^{-2}$;

- Weak mixing angle: $\sin^2 \theta_W = 0.22262$, calculated from $\sin^2 \theta_W = 1 - \frac{m_W^2}{m_Z^2}$;

- Fine-structure constant: $\alpha = \frac{1}{132.3384}$, derived from $\alpha = \frac{\sqrt{2}}{\pi} m_W^2 G_F \left(1 - \frac{m_W^2}{m_Z^2}\right)$.

The five-flavor scheme is used, with the bottom quark treated as massless ($m_b = 0\,\text{GeV}$). The parton distribution function (PDF) set used across all generators is NNPDF30_NLO_AS_01180, accessed via the LHAPDF interface [56]. The values of the strong coupling $\alpha_s$ are obtained from the PDF set. The simulations are performed at a center-of-mass energy of $\sqrt{s} = 13\,\text{TeV}$, with the renormalization and factorization scales set to the central scale choice $\mu_0 = \mu_R = \mu_F = m_{4\ell}/2$, where $m_{4\ell}$ is the invariant mass of the four leptons. Since we are mainly interested in the description of QCD radiation at the matrix element level and during the subsequent parton shower, we disable hadronization, multi-parton interactions, and QED radiation in both PYTHIA8 and SHERPA. In our analysis, jet reconstruction is performed by clustering all stable particles using the anti-$k_t$ algorithm [57] with a radius parameter of $R = 0.4$. Only jets with $p_T^j > 20\,\text{GeV}$ are considered. To avoid the region where the Higgs boson is produced on-shell, the invariant mass of the four leptons is restricted to $150\,\text{GeV} < m_{4\ell} < 500\,\text{GeV}$. The range reflects the most experimentally relevant region where the differential cross-section remains significant and well-measured. Additionally, cuts are applied to the invariant mass of same-flavor lepton pairs: $60\,\text{GeV} < m_{\ell\ell} < 120\,\text{GeV}$, isolating the $Z$ peak.

## 3 Differential Cross-Sections with QCD Effects

We begin by performing a detailed comparison of the fixed-order LO results from POWHEG, SHERPA, and MADGRAPH. This step is crucial to ensure consistent parameter settings across the generators. This way, any discrepancies observed in subsequent comparisons, which will

include higher-order QCD effects, can be attributed to the physics of interest. The LO cross-sections for the signal, background, and inclusive contributions are presented in Table 1. All results are consistent within one sigma. We also compared the differential distribution of the four-lepton invariant mass ($m_{4\ell}$) and observed good agreement.

| Process | POWHEG LO | SHERPA LO | MADGRAPH LO |
|---|---|---|---|
| Signal | 0.08745(5) | 0.08741(5) | 0.08742(1) |
| Background | 2.725(1) | 2.726(1) | 2.724(1) |
| SBI | 2.617(1) | 2.616(1) | 2.617(1) |

Table 1: Comparison of fixed-order LO cross-sections (in fb) for signal, background, and inclusive processes as calculated by POWHEG, SHERPA, and MADGRAPH.

We extend our comparison to evaluate higher-order QCD effects by including results from NLO calculations (POWHEG) and the 0+1 jet merging procedure (SHERPA and MADGRAPH), all matched to parton shower as described in Sec. 2.1. We begin with the fiducial cross-sections, presented in Table 2. Looking at the NLO values, we observe a substantial increase compared to LO, with $k$-factors ranging between 1.7 for the background and SBI to 2.0 for the signal, which is typical of color singlet production through gluon fusion [58, 59]. On the contrary, the LO(0+1j) cross-sections from SHERPA and MADGRAPH increase mildly for the signal and *decrease* for the background and SBI. This indicates that a substantial contribution to the NLO corrections originates from either virtual corrections or unresolved real radiation, both of which are omitted by jet-merging methods but included in the NLO description. We will return to this point in our discussion of the transverse momentum spectra in Sec. 3.

| Process | POWHEG NLO+PS | SHERPA LO(0+1j)+PS | MADGRAPH LO(0+1j)+PS |
|---|---|---|---|
| Signal | 0.1759(5) | 0.09300(4) | 0.09125(8) |
| Background | 4.74(1) | 2.102(1) | 1.914(4) |
| SBI | 4.49(1) | 2.005(1) | 1.892(3) |

Table 2: Comparison of cross-sections (in fb) considering QCD effects beyond fixed-order LO for POWHEG (NLO+PS), and for SHERPA and MADGRAPH (LO(0+1j)+PS), showing the large increases from LO to NLO.

We turn now to differential distributions, beginning with the invariant mass of the four-lepton system $m_{4\ell}$, which we show in Figure 1a for the signal and in Figure 1b for the SBI processes. The results for the background process, for this and all observables that we studied, are very similar to those for SBI, and we display them in App. A. The first ratio plot displays the ratio of the POWHEG distributions (at NLO+PS) to those from SHERPA and MADGRAPH (at LO(0+1j)+PS). This ratio can be taken as a partial indication of the magnitude of the missing NLO effects in the jet-merged samples, although other differences between the merging and matching procedures are also reflected. The second ratio plot illustrates theoretical uncertainty bands from scale variations, where both the renormalization ($\mu_R$) and factorization ($\mu_F$) scales were varied by taking $\mu_R = \mu_F = 0.5\mu_0$ and $\mu_R = \mu_F = 2.0\mu_0$, with $\mu_0$ being the default scale choice. As expected, the scale dependence is smaller for NLO results compared to LO, with POWHEG showing an uncertainty of approximately 18% for the signal, compared to around 27% for MADGRAPH and about 43% for SHERPA. The scale uncertainties are slightly smaller for the SBI results, but the hierarchy remains similar. As expected for an inclusive observable, the shape of the invariant mass distribution is very similar across the three generators. For the signal, the ratio plot fluctuates by about 5% around a central value of 2, which aligns with the cross-section values shown in Table 2. The situation is similar for the SBI re-

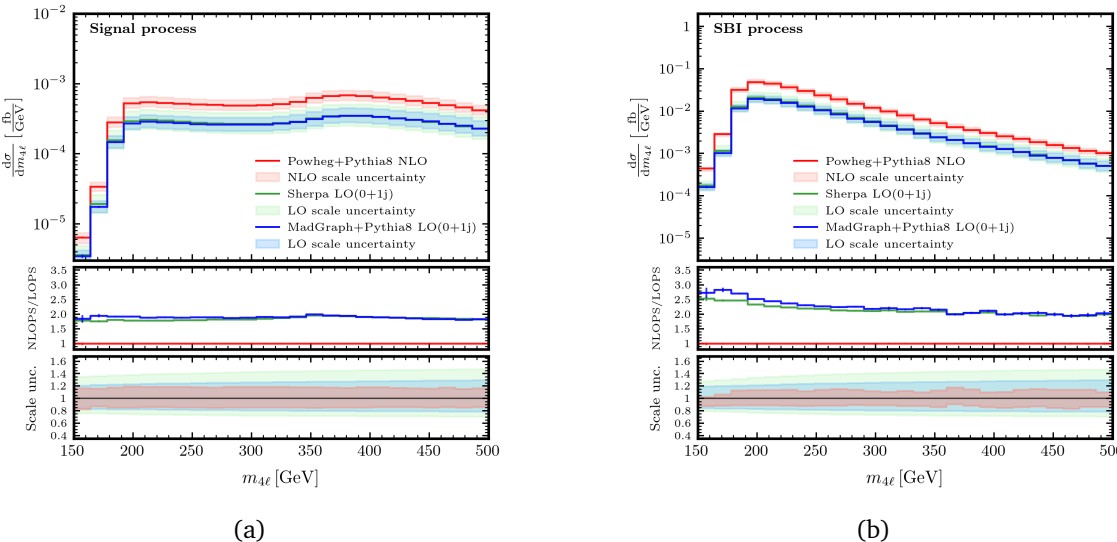

(a)             (b)

Figure 1: Comparison of differential cross-sections predicted by POWHEG (red), MAD-GRAPH (blue), and SHERPA (green) as a function of the invariant mass of the four-lepton system, $m_{4\ell}$, for the (a) signal and (b) SBI processes. The middle panes display the ratio of the MADGRAPH and SHERPA results to those from POWHEG , while the bottom panes illustrate the uncertainty band from renormalization and factorization scale variations, following the same color code as in the nominal predictions.

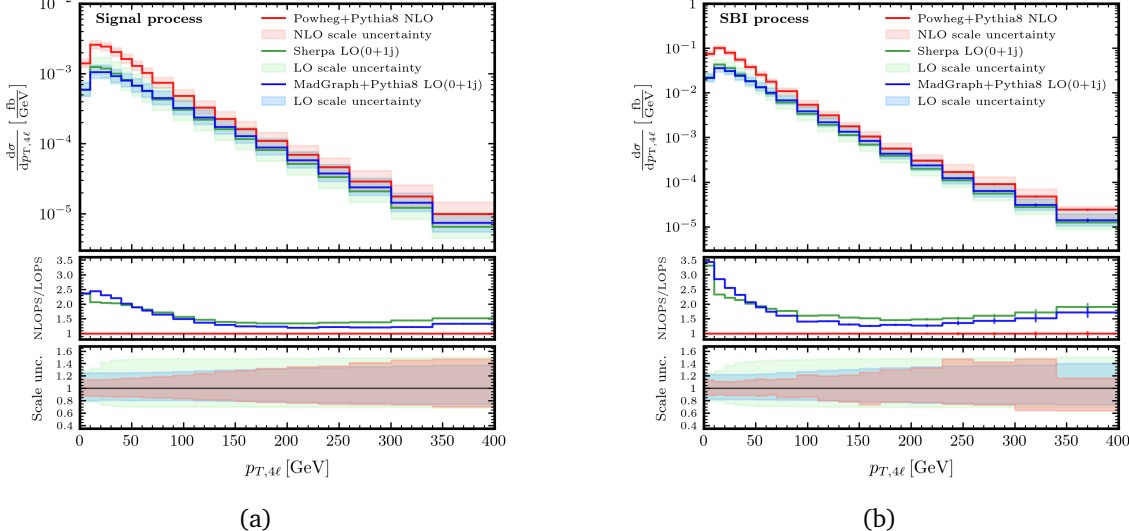

(a)             (b)

Figure 2: Comparison of differential cross-sections predicted by POWHEG (red), MAD-GRAPH (blue), and SHERPA (green) as a function of the transverse momentum of the four-lepton system, $p_{T,4\ell}$, for the (a) signal and (b) SBI processes. The middle panes display the ratio of the MADGRAPH and SHERPA results to those from POWHEG, while the bottom panes illustrate the uncertainty band from renormalization and factorization scale variations, following the same color code as in the nominal predictions.

sults, although the fluctuations are somewhat larger, at around 15% and we observe a slight increase in this ratio at lower values of $m_{4\ell}$, which is likely due to the different treatment of soft radiation close to the $2m_Z$ threshold.

In contrast, the transverse momentum $p_{T,4\ell}$ (shown in Figure 2) is a more exclusive observ-

able sensitive to additional radiation, and it shows a pronounced difference in shapes obtained using NLO+PS matching and jet merging. The POWHEG results at low transverse momentum are enhanced by the large virtual and unresolved-real radiation corrections (both of which are absent from the results using jet-merging), which are subsequently spread out over finite transverse momentum values by the parton shower. As a consequence, the POWHEG spectrum is significantly softer than the distributions obtained using MADGRAPH and SHERPA. The agreement is somewhat better at high transverse momentum, a region particularly relevant for measurements in the $gg \rightarrow H \rightarrow ZZ \rightarrow 2\ell 2\nu$ channel, although here the matched and merged results still differ by factor of around 1.5, growing slightly with $p_{T,4\ell}$ for SBI. We recall that this observable reflects the total transverse momentum of all QCD radiation against which the four-lepton system recoils. As a result, the difference in the high $p_{T,4\ell}$ tail is likely due to an interplay of various effects, including the treatment of additional radiation in the merging and matching approaches (e.g. the scale at which $\alpha_s$ is evaluated) and their interface with the parton shower. We also observe a notable difference of around 15% between MADGRAPH and SHERPA, which could be due to differences in merging schemes (MLM vs. CKKW) as well as the parton shower. Finally, we note that the scale uncertainties across the bulk of the $p_{T,4\ell}$ spectrum are similar for each of the three generators, reflecting the fact that POWHEG only provides LO control on this observable.

As mentioned above, the transverse momentum $p_{T,4\ell}$ is sensitive to the sum of all QCD radiation, so it is also interesting to compare results for the leading and sub-leading jets. These are shown in Figure 3. As anticipated, the results for the leading jet, $p_{T,j_1}$, are almost identical to those shown in Figure 2 for $p_{T,4\ell}$, indicating that the differences between the NLO+PS matched and jet-merged results shown in that figure are indeed largely driven by the presence of the virtual corrections in the former, rather than by additional radiation provided by the parton shower. The results from the three generators still differ in the high-$p_{T,j_1}$ tail, although less so than at large values of $p_{T,4\ell}$. As before, this residual difference could be caused by the treatment of radiation in the matching and merging approaches. We also note that the MADGRAPH and SHERPA results are almost identical, as expected since this observable should be largely insensitive to the details of the merging procedure and the parton shower. There are more noteworthy differences in the distributions of the sub-leading jet $p_{T,j_2}$. SHERPA predicts a somewhat harder spectrum compared to POWHEG, while MADGRAPH severely underpopulates the high-$p_{T,j_2}$ tail. We are unable to explain this latter observation, since the sub-leading jet is generated by PYTHIA8, used both by POWHEG and MADGRAPH. We conjecture that the deficit is related to the details of the MLM matching procedure, but the dedicated studies that would be required to confirm this are beyond the scope of this work. We further note that understanding the kinematics of the sub-leading jet is especially relevant for experimental measurements which identify electroweak Higgs boson production through the observation of jet pairs with a large rapidity gap, as is the case for the most recent ATLAS and CMS measurements of off-shell Higgs boson production.

Many of the above observations can be summarized through the jet-binned cross-sections, shown in Figure 4. The POWHEG results provide larger and more perturbatively accurate results in the zero-jet bin, compared to MADGRAPH and SHERPA. The agreement is much better in the one-jet bin, although the POWHEG rates are still around a factor of 1.5 larger than MAD-GRAPH or SHERPA. As more jets are included by the parton shower, the agreement between POWHEG and SHERPA improves, with MADGRAPH generating substantially fewer jets.

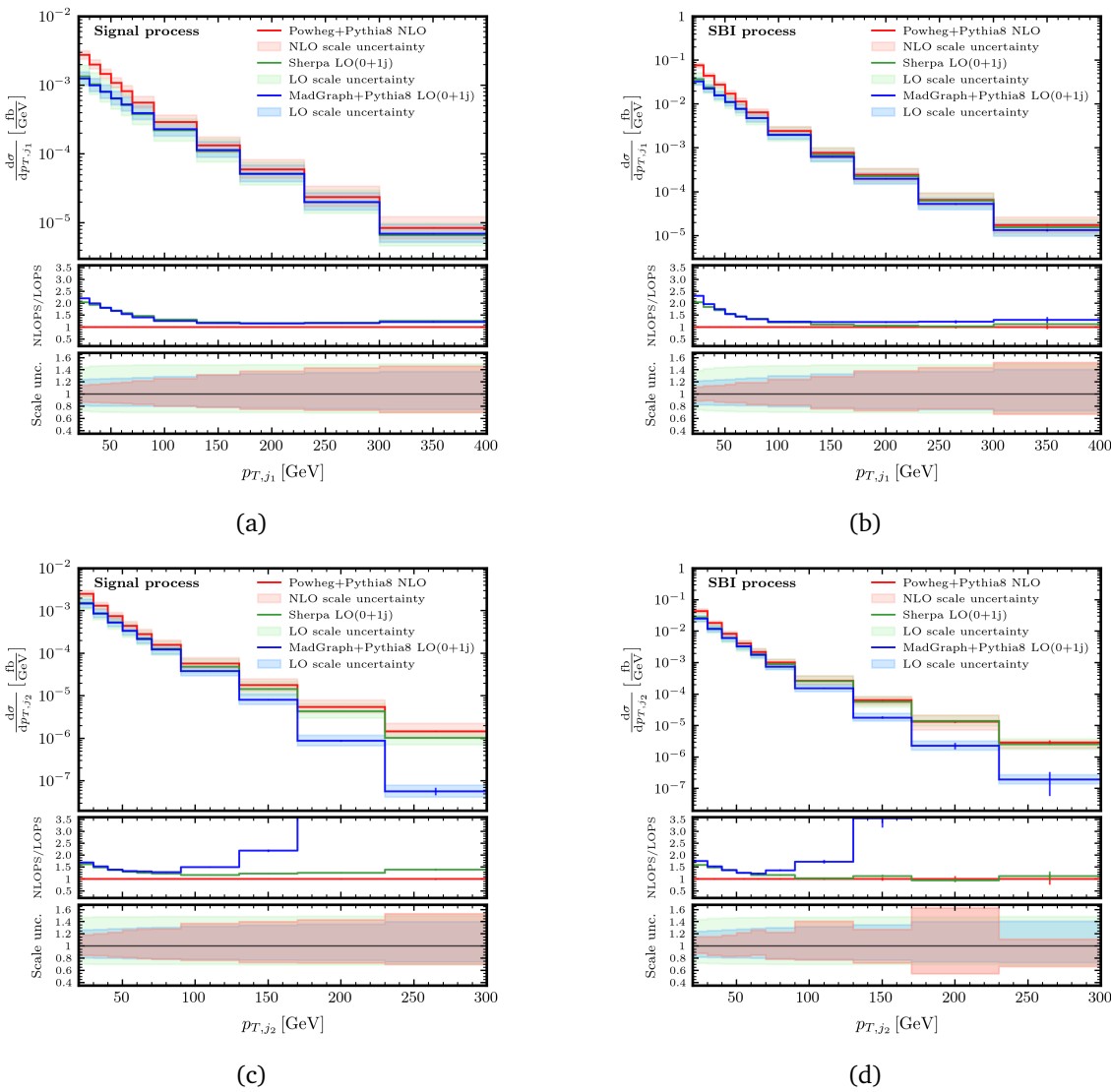

Figure 3: Comparison of differential cross-sections predicted by POWHEG (red), MAD-GRAPH (blue), and SHERPA (green) as a function of the transverse momentum of the leading jet $p_{T,j_1}$ for the (a) signal and (b) SBI processes, and corresponding comparisons (c, d) as a function of the sub-leading jet $p_{T,j_2}$. The middle panes display the ratio of the MADGRAPH and SHERPA results to those from POWHEG, while the bottom panes illustrate the uncertainty band from renormalization and factorization scale variations, following the same color code as in the nominal predictions.

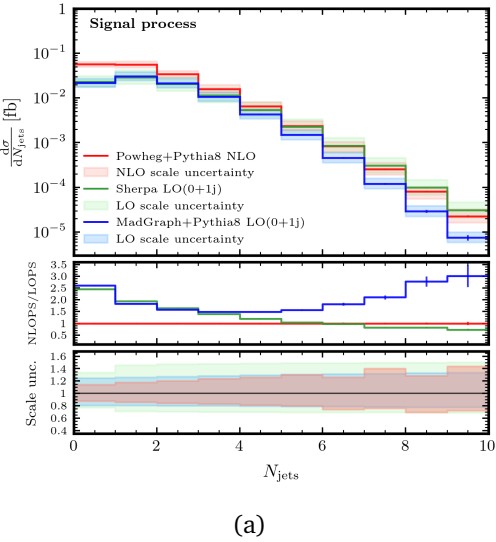

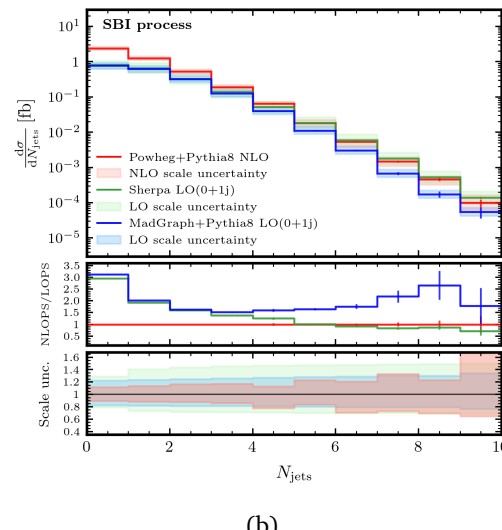

(a)

(b)

Figure 4: Comparison of differential cross-sections predicted by POWHEG (red), MAD-GRAPH (blue), and SHERPA (green) as a function of the jet multiplicity, $N_{\text{jets}}$, for the (a) signal and (b) SBI processes. The middle panes display the ratio of the MAD-GRAPH and SHERPA results to those from POWHEG, while the bottom panes illustrate the uncertainty band from renormalization and factorization scale variations, following the same color code as in the nominal predictions.

## 4 Theoretical Uncertainty Impact

Accurately assessing the uncertainties associated with the theoretical modeling is extremely important, especially when comparing Monte Carlo generators with different underlying formalisms, such as parton shower matching and jet merging. Ideally, one would do so by closely examining the algorithms used to generate partonic radiation, but this is extremely demanding. As proxies, one can estimate the uncertainties by varying key parameters that affect event generation and/or by considering the differences between results produced by different MC codes. We present results using the former option in this section.

In the POWHEG NLO matching procedure, the parameter $h_{\text{damp}}$ regulates the amount of radiation that gets exponentiated by the Sudakov factor. The results of the previous section were obtained using a nominal value of $h_{\text{damp}} = 100$ GeV, following Ref. [24, 60], and we now consider variations of $h_{\text{damp}} = 75$ GeV and $h_{\text{damp}} = 150$ GeV. Additionally, we vary the renormalization scale controlling initial-state radiation (ISR) relative to its nominal value $\mu_0$ ($\mu_R^{\text{ISR}} = 2\mu_0$ or $\mu_R^{\text{ISR}} = 0.5\mu_0$), in order to assess the sensitivity of radiation patterns to ISR modeling. The merging procedure in SHERPA and MADGRAPH introduces the parameter $Q_{\text{cut}}$ which defines the transition between jets generated by the matrix element and those generated by the parton shower. Its nominal value is set to 20 GeV, which we vary to 15 GeV and 30 GeV. MADGRAPH also introduces the parameter `alpsfact`, which scales the strong coupling constant at the initial parton shower scale. The nominal value is 1.0, and we now consider values of 0.5 and 2.0. SHERPA also introduces the parameters `QSF` and `CSSKIN`, which are critical for controlling parton shower behavior. The former acts as a scaling factor for the renormalization and factorization scales used in the parton shower, directly influencing the probability of soft and collinear emissions, while the latter modifies the Catani–Seymour splitting kernel, impacting the Sudakov factor and adjusting the kinematic constraints on parton splitting during the showering process. In all simulations, the value of the `QSF` scale used was

the same as the renormalization scale $m_{4\ell}/2$, and it was varied between half and twice this value to assess the uncertainty. The nominal simulation used the recoil scheme described in Ref. [50], while the variation considered the alternative approach from Ref. [61][4]. Note that the CSSKIN variation affects only the second and subsequent shower emissions.

Figure 5 presents the normalized differential cross-sections of $m_{4\ell}$ and $p_{T,4\ell}$ for the signal and SBI processes. The first ratio plot displays the ratio with respect to the POWHEG results (i.e., NLO+PS), using the nominal values of all parameters, and serves to highlight the shape differences that we discussed in the previous section. The second ratio plot shows theoretical uncertainty bands, obtained by summing the variations described above in quadrature. Among all parameter variations, QSF uncertainty in SHERPA has the largest impact across the entire $m_{4\ell}$ range and at low $p_{T,4\ell}$ values, with uncertainties reaching up to 40%-50%. In contrast, $h_{\text{damp}}$ uncertainty in POWHEG is prominent primarily in $p_{T,4\ell}$, especially in the range from about 130 GeV to about 230 GeV, affecting the signal process with uncertainties up to 20%. For higher $p_{T,4\ell}$ values, the CSSKIN parameter in SHERPA becomes the most significant source of uncertainty. The MADGRAPH uncertainties for this observable are largest at smaller transverse momenta, reaching around 15% at $p_{T,4\ell} \approx 40$ GeV.

We also note that the uncertainties obtained from varying the parameters are larger than the differences between the results produced by the three programs for the $m_{4\ell}$ distribution, and smaller for the $p_{T,4\ell}$ distribution. This suggests that the uncertainty obtained by varying parameters is fairly conservative for the $m_{4\ell}$ distribution, but possibly an underestimate for the $p_{T,4\ell}$ distribution.

From these comparisons, the advantage of using the NLO accuracy of POWHEG is most clear for the inclusive observable $m_{4\ell}$. On the other hand, for an exclusive variable like the transverse momentum of the four-lepton system, determining the most effective generator is less obvious, since all three have LO accuracy, and the uncertainties obtained from varying parameters in MADGRAPH and POWHEG are comparable.

## 5    Conclusion

We have presented the first comprehensive comparison of jet-merged and NLO+PS predictions for off-shell Higgs boson production, emphasizing the impact of higher-order QCD effects and theoretical uncertainties. We observe very similar shapes from POWHEG, MADGRAPH, and SHERPA for the invariant mass $m_{4\ell}$ system, although POWHEG displays smaller uncertainties, as anticipated. On the contrary, the modeling of additional radiation leads to different shapes for the transverse momentum distributions, with the POWHEG results being systematically softer than those obtained using merging programs. We have interpreted this effect as being caused by the large NLO corrections at low $p_{T,4\ell}$ which are then spread out by the parton shower. We also observed that MADGRAPH underpopulates the sub-leading jet distribution at transverse momenta above roughly 80 GeV, compared to the other two generators studied. Similarly MADGRAPH predicts fewer events with three or more resolved jets.

We also studied the uncertainties associated with each of the generators, observing that the uncertainties from POWHEG are the lowest for the $m_{4\ell}$ distribution, while those of POWHEG and MADGRAPH are comparable for the $p_{T,4\ell}$ spectrum. The uncertainty associated with SHERPA appears to be the largest for both the observables studied. Thus the NLO+PS generator provided by POWHEG appears to give the most robust theoretical predictions for off-shell Higgs boson studies. However, as already observed in Ref. [24], the impact of the parton shower in POWHEG at large $p_{T,4\ell}$ is quite large, due to the choice of the default recoil scheme in PYTHIA8. Additionally, the uncertainty estimated from the variation of the damping parameter can be a

---

[4]The change is obtained by varying the CSS_KIN_SCHEME parameter from 1 (default) to 0 (alternative).

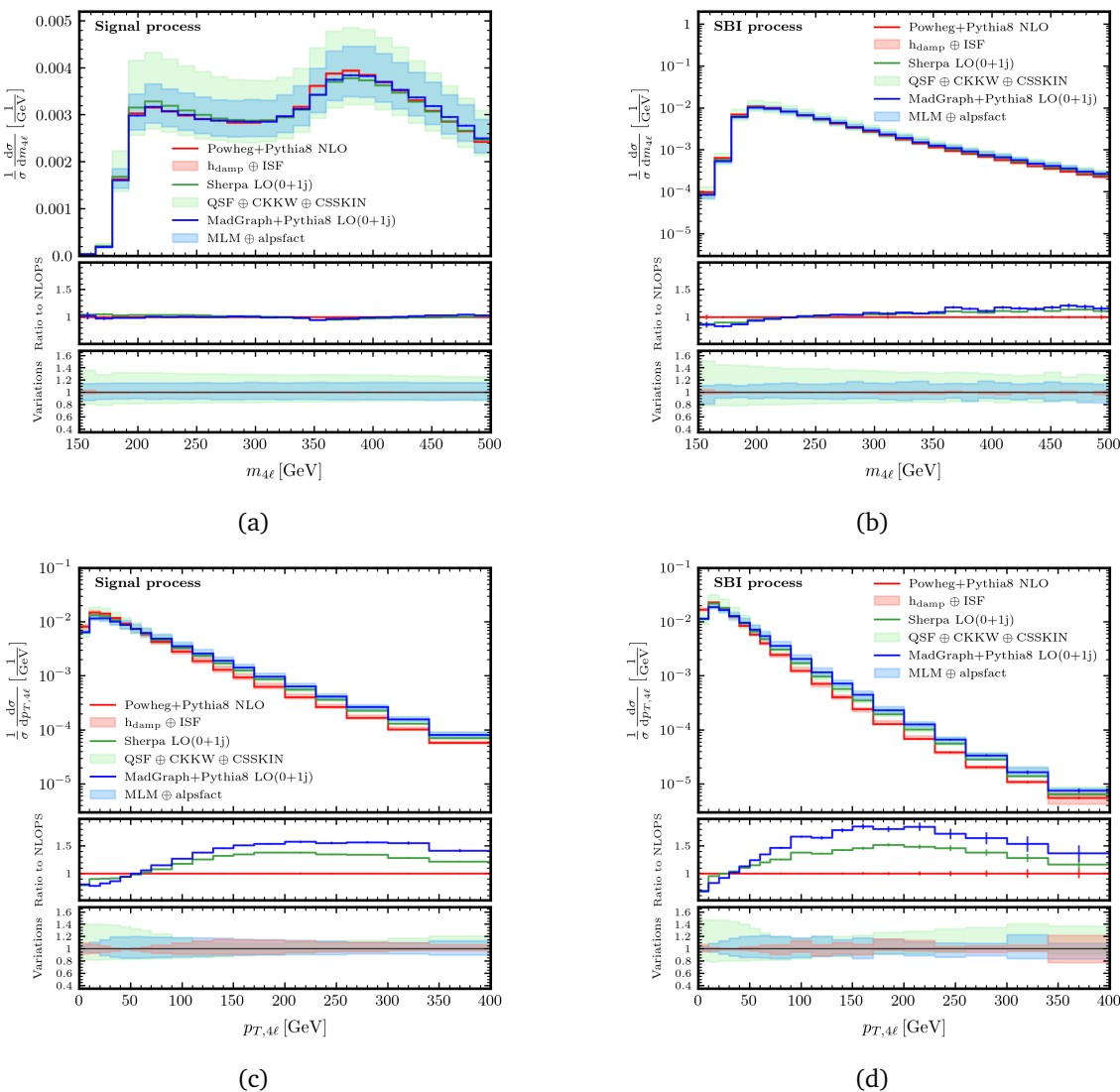

Figure 5: Comparison of differential cross-sections predicted by POWHEG (red), MAD-GRAPH (blue), and SHERPA (green) as a function of the invariant mass of the four-lepton system $m_{4\ell}$ for the (a) signal and (b) SBI processes, and corresponding comparisons (c, d) as a function of the transverse momentum of the four-lepton system $p_{T,4\ell}$. The middle panes display the ratio of the MADGRAPH and SHERPA results to those from POWHEG, while the bottom panes illustrate the uncertainty band from renormalization and factorization scale variations, following the same color code as in the nominal predictions. The uncertainty in the MLM and CKKW jet-merging algorithms are estimated by varying the $Q_{\text{cut}}$ scale.

substantial source of modeling uncertainty in future measurements. We note that the uncertainties discussed here are just a part of a complete model and that other uncertainties, such as those arising from the choice of top quark mass scheme [62], could be included in future measurements.

Promising future studies along similar lines include comparing results at NLO+PS accuracy with 0+1+2j merged samples, or comparisons with detailed experimental measurements of the $gg \to ZZ$ process. Nevertheless, we believe that this study offers valuable insights into the precision of theoretical models and their potential for improving off-shell Higgs investigations, including indirect measurements of the Higgs boson width.

# Acknowledgements

This work has been carried out within the LHC Higgs Working Group, as a contribution to Report 5. We thank the members of the working group for various discussions on this topic, and for pushing us to pursue the work presented here. We are grateful to Olivier Mattelaer for help with running the MADGRAPH code. RR is partially supported by the Italian Ministry of Universities and Research (MUR) through grant PRIN2022BCXSW9. The work of MJ and RCLSA is supported in part by the US Department of Energy grant DE-SC0010004. This work utilized computational resources from the University of Massachusetts Amherst Research Computing at the Massachusetts Green High Performance Computing Center.

# A   Differential cross-sections for the background process

In this appendix, we display the differential cross-sections for the background process. Comparing the plots shown in Figure 6 with those in Figures 1, 2, and 3 of the main text, the similarities between the SBI and background distributions are readily apparent.

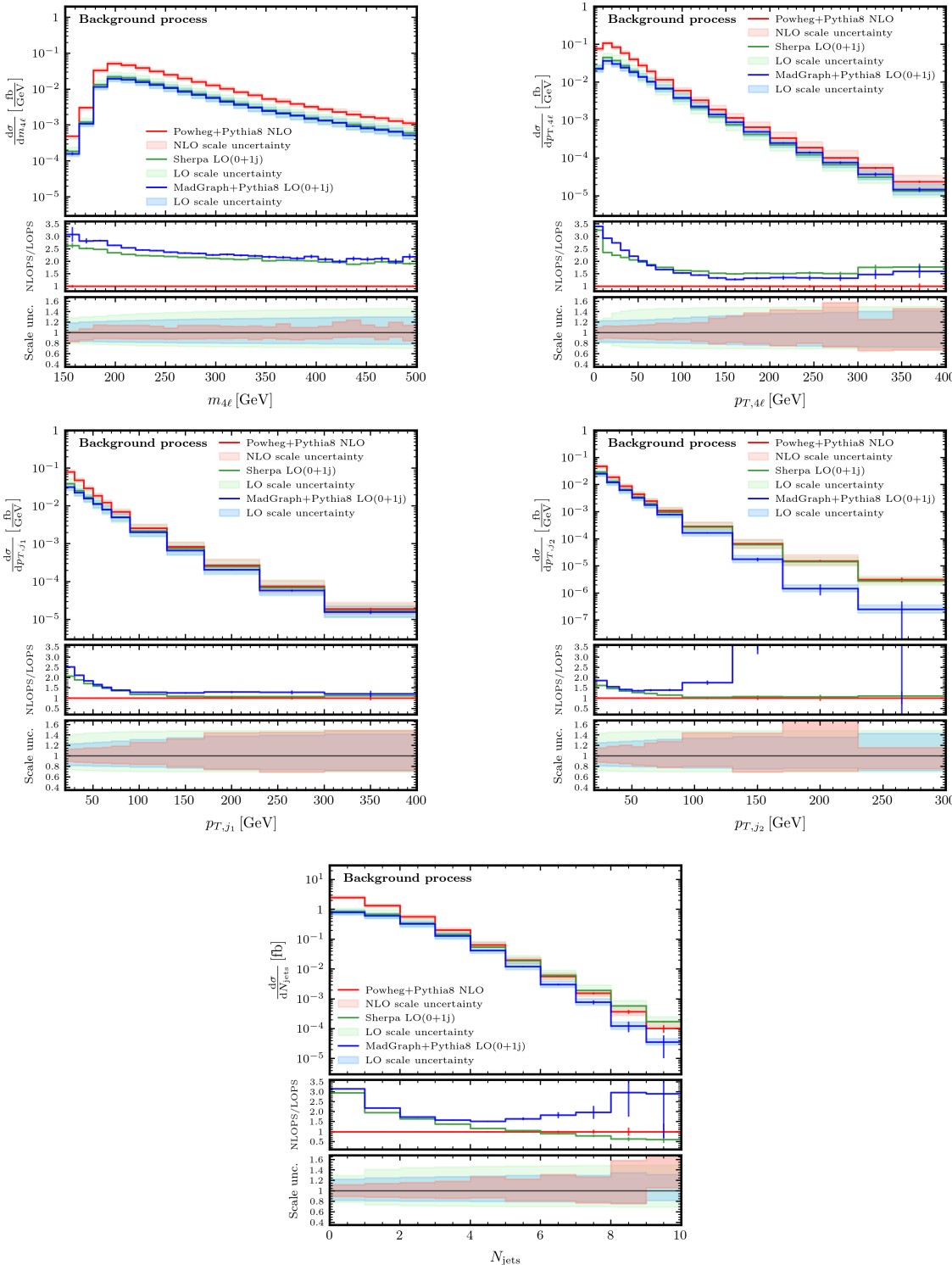

Figure 6: Comparison of differential cross-sections predicted by POWHEG (red), MAD-GRAPH (blue), and SHERPA (green) for the background process. The layout and content follow the same structure as in the main text (see Figure 1).

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
