# Peer review of "Theoretical modeling of QCD radiation in off-shell Higgs production through gluon fusion"

_SciPost Physics Community Reports_

## Round 1 · Referee Report · Anonymous (Referee 1) · 2025-8-5

Strengths

1- The uncertainties and level of agreement between three available event-generation frameworks is characterised, allowing simulation setups to make an informed choice of software.

Weaknesses

1- A discrepancy between generators for one of the distributions (ptj2) is not fully understood, and deferred to future work.

Report

This manuscript makes a detailed study of theory uncertainties for modelling off-shell Higgs production for LHC physics studies, important for studies of the Higgs width. These uncertainties will be particularly relevant for the HL-LHC, where statistical uncertainties will no longer be the dominant ones.

The work compares total and differential rates, modelled by three event-generator setups. One includes NLO accurate cross sections (which are available in the literature) and two use 0+1j merging methods. Common input parameters and settings are used, to enable a comparison of the predictions.

The comparisons find that the NLO effects are large, which is in line with other gluon-fusion di-boson processes. The shapes of m4l, pt4l, ptj1 distributions broadly agree, but notably not for ptj2, the transverse momentum of the second hardest jet. In particular the prediction of madgraph+pythia8 is not well understood.

Uncertainties are presented due to variation of scales, and also generator parameters (hdamp, etc). All three setups agree within the uncertainties for the invariant mass distribution, but not for the transverse momentum distribution. As such, it is not so clear what uncertainty should be assigned here. The authors acknowledge that not all sources of uncertainty are considered, for example due to the renormalisation of the top quark mass (which has a large effect for other top-loop-induced processes), and note that such uncertainties could be included in future studies.

The authors tentatively recommend powheg+pythia8 for off-shell Higgs studies, which includes NLO accurate virtual cross sections.

I find the manuscript suitable for publication in Community Reports for YR5.

Recommendation

Publish (easily meets expectations and criteria for this Journal; among top 50%)

---

## Round 1 · Referee Report · Anonymous (Referee 2) · 2025-8-6

Strengths

  1. Valuable comparison of different Monte Carlo generators

Weaknesses

  1. Some discrepancies are not fully investigated

Report

The authors perform a comparison of three different Monte Carlo setups in the $gg \rightarrow ZZ$ channel. The process is of particular interest to the high-energy physics community as it is used by the ATLAS and CMS experiments for the indirect determination of the Higgs boson width. The authors compute the signal arising from an off-shell Higgs boson, the background and the signal-background interference contributions with POWHEG+Pythia at next-to-leading order, Madgraph5+Pythia and SHERPA (both at leading order) and investigate the impact of different parameters relevant for matching. Comparisons between Monte Carlo tools are always valuable to the community, especially in the presence of higher order and radiative corrections where systematic uncertainties could arise. I believe the manuscript fulfils one of the journal’s expectations, by filling a gap concerning reference results or benchmarks in a particular research direction and does have potential for follow-up work. Before recommending its publication, I have a couple of suggestions.

Requested changes

  1. The authors should mention how the K-factors are calculated for the SBI. Do they use the background K-factor or $\sqrt{K_{bkg} K_{signal}}$?
  2. The authors mention that there is a difference between MadGraph and Sherpa which could be due to differences in merging schemes, MLM vs CKKW. Since the latter option is also available in Madgraph, it could be investigated to see if CKKW in Madgraph+Pythia indeed reduces the difference with respect to Sherpa. This could be interesting to check for the sub-leading jet $p_{T, j_2}$.

Recommendation

Ask for minor revision

  • validity: -
  • significance: -
  • originality: -
  • clarity: -
  • formatting: -
  • grammar: -

Author:  Rafael Coelho Lopes de Sa  on 2025-08-11  [id 5718]

(in reply to Report 2 on 2025-08-06)

Dear editor and referee,

We thank you for your careful reading of our paper and evaluations. We have addressed the changes requested as follows:

  1. The k-factors for SBI quoted on p.6 are taken from a simple ratio of the values shown in tables 1 and 2, so that we do not need to use any approximation for the SBI k-factor. To clarify this, we have modified the sentence

"Looking at the NLO values, we observe a substantial increase..."

to

"Looking at the NLO values, and comparing these to the LO value in Table 1, we observe a substantial increase..."

on p.6.

  1. A comparison with results obtained using the CKKW scheme in MadGraph would indeed be very helpful, and indeed we considered obtaining these results. Unfortunately, MadGraph's CKKW-L merging cannot handle loop-induced processes, as noted in e.g. the ATLAS MadGraph manual (https://gitlab.cern.ch/atlas/athena/-/tree/main/Generators/MadGraphControl) as well as public talks on similar simulations (see e.g. slide 11 of the talk at https://indico.cern.ch/event/966917/contributions/4072035/attachments/2128200/3583798/2jet_HXSWG_21Oct.pdf).

To clarify this for the reader, we have added a footnote on p.4.

"We note that the CKKW-L scheme is not available in MadGraph for loop-induced processes, such as those considered in this paper."

With these minor changes, we hope that the paper is ready for publication. Following the instructions from SciPost, a new version with only the two modifications has been submitted to the arXiv and should appear tomorrow.

---

## Editorial Decision

resubmitted